# Visual Indicator for Intradialytic Hypotension Prediction Using Variation and Compensation of Heart Rate

**DOI:** 10.3390/diagnostics14232664

**Published:** 2024-11-26

**Authors:** Tae-Wuk Bae, Ji-Hyun Park, Jong-Won Park, Kee-Koo Kwon, Kwang-Yong Kim

**Affiliations:** 1Daegu-Gyeongbuk Research Center, Electronics and Telecommunications Research Institute, Daegu 42994, Republic of Korea; kwonkk@etri.re.kr (K.-K.K.); kimky@etri.re.kr (K.-Y.K.); 2Division of Nephrology, Department of Internal Medicine, College of Medicine, Yeungnam University, Daegu 42415, Republic of Korea; karismatajongwon@gmail.com

**Keywords:** intradialytic hypotension, perceptron, heart rate, hemodialysis, real-time

## Abstract

Background: To date, most intradialytic hypotension (IDH) studies have proposed technologies to comprehensively predict the occurrence of IDH using the patient’s baseline information and ultrafiltration (UF) information, but it is difficult to apply the technologies while identifying the patient’s condition in real time. Methods: In this study, we propose an IDH indicator that uses heart rate (HR) change information to identify the patient’s condition in real time and visually shows whether IDH has occurred. The data used were collected from 40 dialysis patients in a clinical trial conducted in the Artificial Kidney Unit at Yeungnam University Medical Center, Korea, from 18 July to 29 November 2023. Results: The IDH indicator infers changes in the patient’s blood pressure during dialysis by analyzing the upper and lower maximum HRs based on the real-time average HR. Medical staff can respond to IDH in real time by looking at the IDH indicator, which visually expresses changes in the patient’s HR. In addition, we propose a multilayer perceptron structure that inputs upper and lower maximum HR information based on the average HR for the time interval accumulated in real time. In learning using 40 min of data up to 5 min before IDH occurred, models using two and five layers showed excellent performance, with accuracy of 88.6% and 85.2%, respectively. Conclusions: By combining IDH visual indicators and the multi-layer perceptron method, medical staff can effectively respond to IDH in real time.

## 1. Introduction

Intradialytic hypotension (IDH) is a frequent side effect during dialysis and is difficult to predict, exposing many patients to the risk. The incidence of IDH during dialysis treatment is approximately 15~20% [1,2]. Typically, a decrease in blood pressure of more than 20 mmHg relative to systolic blood pressure (or more than 10 mmHg relative to mean arterial pressure) is referred to as IDH [3,4]. It is reported that frequent occurrence of IDH is a factor that increases mortality [5,6] and can cause various risks to the heart and central nervous system [7,8].

The primary cause of IDH is the reduction in blood volume, which occurs when a large amount of fluid is removed from the vascular compartment during ultrafiltration (UF) and there is insufficient fluid refilling from the interstitial compartment to the vascular compartment [9]. First, the causes of low blood pressure due to decreased blood volume are as follows [10,11]: (1) when the UF amount exceeds the plasma recharge amount, (2) when the dialysis time after weight gain is short, (3) when UF is performed below the dry weight, and (4) when the Na concentration of the dialysate is too low. In the fourth case, if the Na concentration of the dialysate is lower than that of the plasma, the plasma becomes low-concentrated and body fluid leaks out of the blood vessels. Excessive UF can reduce cardiac output and increase the likelihood of developing IDH, especially when compensatory mechanisms such as myocardial contraction and heart rate (HR) regulation do not function optimally [5]. Second, the causes of low blood pressure due to lack of blood vessel constriction are as follows [12]. When blood volume is reduced, constricting blood vessels is very important to maintain blood pressure. (1) Antihypertensive drugs, (2) high dialysate temperature, and (3) acetate dialysate may cause vasodilation.

The body’s compensatory mechanisms for intravascular hypovolemia include recharge of plasma from interstitial and intracellular compartments, the heart’s response to maintain cardiac output and venous circulation, and arteriolar vasoconstriction to increase total peripheral resistance [13,14]. When IDH occurred, HR and peripheral resistance increased in patients with normal autonomic function, whereas in patients with impaired autonomic function, total peripheral resistance decreased even under a fixed HR [15]. To address IDH, it is essential to reduce UF pressure and blood flow rate in order to slow down the decrease in blood volume [9]. The second step is to administer physiological saline, high-concentration saline, or high-concentration glucose to increase blood pressure [16]. The third step is to increase the Na concentration of the dialysate. Since these practical measures are taken when symptoms such as low blood pressure and muscle cramps appear to some extent, medical staff need some technology to determine the patient’s condition and IDH in real time during dialysis.

The currently proposed IDH predictors or indicators have used heart rate variability (HRV) [17], UF rate control based on blood volume [18,19], and fluid status assessment based on B-line count [20] as representative factors for predicting IDH. In [17], it was found that patients who developed IDH had hypertension at the start of dialysis, and their HRV indices decreased as the dialysis progressed. However, to ensure the reliability of the result, more HRV measurements and corresponding blood pressure measurements during dialysis are required, but obtaining such data during dialysis is practically challenging. In [18], IDH was found to be caused by hypovolemia resulting from UF, and it was reported that blood volume (BV)-controlled UF contributes to intradialytic stability. On the other hand, ref. [19] reported that dialysis using BV-controlled UF control did not reduce the prevalence of IDH compared to standard dialysis, contrary to the findings of [18]. In [20], it was reported that arrhythmias, low pre-dialysis systolic blood pressure, and a low number of B-lines on lung ultrasound were associated with a patient’s fluid status and the development of IDH. However, due to the small number of clinical patients, it remains unclear whether the routine use of lung ultrasound is effective in preventing IDH.

Recently, many studies based on bio-signal analysis have been conducted to predict the onset of IDH. IDH prediction methods using hemodynamic instability based on short-term fluctuations in oxygen saturation [21], amplitude, envelope, and frequency analysis [22,23,24] of photoplethysmography reflecting vasoconstriction and cardiac output, and autonomic neuropathy [25] have been proposed. In addition to these single-variable-based methods, multivariate-based machine learning or deep learning methods have recently been proposed to predict IDH, e.g., a multivariate negative binomial model using patient information and HRV [13], a multivariate regression model based on age [26], a time-dependent logistic regression model based on systolic blood pressure and demographic variables [27], multivariate regression model based on UF and underlying disease [28], and a deep learning model based on patient baseline information and HR slope [29]. However, although these mentioned methods have high accuracy in predicting IDH, they are difficult to apply in real time because they do not utilize real-time bio-signal characteristics and are not practical for medical staff to use. Therefore, this paper proposes a HR-variation-based IDH indicator and a multilayer perceptron model that allow medical staff to monitor the patient’s condition in real-time. It was assumed that IDH occurs due to intravascular hypovolemia caused by a decrease in body fluid during dialysis. Blood pressure lowered by IDH causes a decrease in HR, and a rapid increase in HR was discovered as a compensation mechanism to raise the lowered blood pressure. In other words, this paper infers blood pressure changes and IDH through HR changes and proposes a system that informs medical staff of this information in real time.

## 2. Materials and Methods

### 2.1. Correlation Between HR and Blood Pressure

Figure 1 shows the symptoms of hypotension that occur during hemodialysis and the physiological compensatory response to it. Reduction in intravascular blood volume due to UF, desensitization to vasopressin (vasoconstrictor hormone), and deactivation of the sympathetic nervous system can lead to a decrease in cardiac output and total peripheral resistance, resulting in hypotension [30]. As a compensatory response to this decrease in blood pressure, a momentary increase in HR (or HR fluctuation) may occur to increase blood pressure. Such changes in HR during dialysis can be used as a clue to understand the occurrence of IDH.

Pulse transit time (PTT) is the time it takes for a pulse wave to travel between two points in the cardiovascular system, measured using the peak-to-peak (or foot-to-foot) method of the pulse wave. The correlation between PTT and blood pressure has been demonstrated in numerous studies [31,32,33], and it is commonly used as an indicator for estimating blood pressure. Generally, when blood pressure is high, blood moves more quickly, resulting in a shorter PTT over the same distance, whereas when blood pressure is low, blood moves more slowly, leading to a longer PTT over the same distance. Thus, PTT and blood pressure are known to be inversely related, assuming that blood density is the same for the same individual [34,35]. Meanwhile, in an HRV study using the P-P interval of the pulse wave and the R-R interval of the ECG, the largest error among the HRV indicators was only 2.46% [36,37]. Building on these existing research findings, this study aimed to predict the occurrence of IDH through the analysis of macro-level HRV using electrocardiography (ECG).

### 2.2. Participants and Devices

The data used for IDH analysis were collected clinically from 40 hemodialysis patients at Yeungnam University Medical Center in Daegu, Republic of Korea (IRB File No.: YUMC 2022-11-025-001). The clinical study was conducted from 18 July to 29 November 2023, and multifunctional measuring devices from Tribell-Lab (Gyeongsan-si, Gyeongsangbuk-do, Republic of Korea), specifically the VDR-1000 [38] and VMA-1000 [39], were used for the correlation study between HR and blood pressure in hemodialysis patients. A total of 120 measurements of biometric signal data, including blood pressure, pulse, and electrocardiography, were taken and stored on a server. As shown in Table 1, there were no significant differences in age, diabetes, hemoglobin levels, or UF rates based on the occurrence of IDH. In this study, IDH was defined as occurring if systolic blood pressure decreased by more than 20 mmHg even once during dialysis. Additionally, while there were more female patients in the IDH occurrence group, this finding was considered clinically insignificant.

### 2.3. Used Data and Visual Marker

To analyze the characteristics of the HR signal of dialysis patients prior to the occurrence of IDH, a 40 min HR signal up to 5 min before the occurrence of IDH was used. In other words, starting from 45 min before IDH occurrence, the signal was incremented in 5 min intervals (300 points, 1 point/s), resulting in HR analysis conducted over a total of 8 time intervals ranging from 5 min to 40 min. Figure 2 displays the HR signals and corresponding IDH indicators for a specific IDH patient, covering the first 5 min (300 points) and the last 40 min (2400 points) up to 5 min before the occurrence of IDH. The original HR signal was too rough for analysis, so an HR curve fitted with a 20th-degree polynomial was used. High HRV, relative to the average HR, indirectly indicates elevated blood pressure, while low HRV suggests reduced blood pressure. In other words, a phenomenon of high HR following a low HR can be seen as the body’s compensatory mechanism to raise lowered blood pressure. The sum of HRs corresponding to the maximum area above and below created by the average HR line and the fitted curve was used as a value representing the HR change in the corresponding time period. In each HR signal, the areas corresponding to the maximum sums of high HRs and low HRs relative to the average HR are indicated by blue and red circles, respectively, and the blue horizontal line in each figure represents the average HR. The left and right sides of the IDH indicator representing HRV display the ratios of the maximum sums of low and high HRs, respectively, indirectly indicating the occurrence rates of low blood pressure and high blood pressure. The maximum values for the sums of each high and low HR were empirically set to 6000 HR changes (5 min = 6000 mmHg/20 mmHg). This value was set to trigger an alert if a decrease of more than 20 mmHg in systolic blood pressure persists for more than 5 min within the 40 min leading up to the occurrence of IDH. In the figure, the IDH indicator starts in a normal state during the initial 5 min and then shows an increase in HR variation and compensatory responses over the 40 min leading up to 5 min before the occurrence of IDH, ultimately providing an IDH warning.

## 3. Results

### 3.1. HR Changes in Indicators for Normal Patients

Figure 3 displays the results of the IDH indicators from the initial 5 min to the last 40 min up to 5 min before the occurrence of IDH for a specific normal patient. Throughout all time intervals from the initial 5 min to 40 min, the high and low HR intervals intersected; however, the amplitude of these HR changes (i.e., the variations in blood pressure) was not significant. As a result, the IDH indicator remained stable in the normal state with an HR change rate of less than 10% until the end of the dialysis session.

Figure 4 presents the IDH indicator results for a representative group of 8 patients among the 53 cases that did not experience IDH. The IDH indicators for these patients show that the percentage of HR variability is below 46%. Unlike patients who experienced IDH, the indicators for normal patients show very low HR variation. In the fitted HR curves of patients with very low HR variability (patients #1, #19, and #42), it can be observed that HR variability remains stable around the average HR. Similarly, in patients #3, #11, #27, and #31, there were continuous temporary HR fluctuations, but the changes in high HR and low HR were evenly mixed around the average HR. In these cases, while temporary blood pressure drops and elevations occur, the likelihood of IDH developing is significantly low. In the case of patient #37, there was a higher HR variability than the average HR at the beginning of the signal, but the overall HR variability was not significant. However, if the maximum sum of low HRs gradually increases after an initial peak in very high HRs, there may be a potential risk of IDH developing, indicating the need for close monitoring.

### 3.2. HR Changes in Indicators for IDH Patients

Figure 5 displays the results of the IDH indicators from the initial 5 min to the last 40 min up to 5 min before the occurrence of IDH for a specific IDH patient. From the initial 5 min to 25 min, there are fluctuations in HR, but since the range of change is not significant, the rate of HR change in the IDH indicator is also relatively small. However, after the 30 min, as the HR increases, the rates of high and low HR change in the IDH indicator become significantly pronounced. This indicates that after a prolonged period of low HR due to low (though not severe) blood pressure up to 30 min, a rapid increase in HR (and consequently blood pressure) occurs as a result of the compensatory mechanisms for blood pressure. This rapid increase in HR is one of the common precursor phenomena observed prior to the occurrence of IDH. Following this rapid increase in HR due to compensatory mechanisms, a significantly lower hypotensive state may occur, increasing the likelihood of IDH. In the 40 min HR signal leading up to 5 min before the occurrence of IDH, prolonged periods of low blood pressure and sudden spikes in high blood pressure are observed, resulting in the maximum rates of high and low HR changes in the IDH indicator, which then provides an IDH warning. When the IDH indicator alerts medical staff about the occurrence of IDH, they can take appropriate measures in advance to prepare for it.

Figure 6 shows the IDH indicator results for 8 representative cases among the 42 IDH cases. It shows that the percentage of HR variation in the IDH indicators for these patients is over 80%. In patients with 100% HR variation (patient #4, #9, #31, and #37), the fitted HR curves show a transition from a period of HR below the average (suspected hypotensive period) to a period of HR above the average (suspected hypertensive period). This can be interpreted as a compensatory mechanism caused by the increase in blood pressure to prevent transient hypotension during dialysis. This compensatory mechanism of blood pressure elevation was a precursor to more severe hypotension, specifically IDH, which later occurred. All of these patients developed IDH within 5 min. This compensatory mechanism of hypertension, namely the elevated HR (HR), can also occur after IDH. This increase in HR occurred similarly in the 8th and 16th patients. In contrast, the 11th, 16th, and 20th patients experienced a shift from a period of elevated HR (suspected hypertensive phase) to a period of low HR (suspected hypotensive phase), and IDH occurred in them 5 min later. These three patients represent cases where transient hypotension immediately progressed to IDH without any compensatory mechanisms to prevent the drop in blood pressure. Additionally, since the maximum sum of the upper HRs in the 8th, 11th, 16th, and 20th patients is greater than the maximum sum of the lower HRs, it indicates a significant compensatory mechanism for blood pressure elevation. In such cases, precautionary measures against IDH should be taken.

### 3.3. Changes in Maximum Area of Upper and Lower HR for Normal and IDH Patients

Figure 7 and Table 2 show the maximum area of upper and lower HRs in 8 time intervals for 53 normal cases and 42 IDH cases. In each figure, the blue, yellow, red, and green lines represent the average curves in that figure. In the case of 53 normal cases, it can be seen that the maximum area of upper and lower HRs for each time section does not exceed 1700 mmHg (approximately 1.41 s = 1700 mmHg/20 mmHg/60 s) and increases monotonically. On the other hand, in the case of 42 IDH cases, it can be observed that the maximum areas of the upper and lower HRs sharply increase from the initial 5 min to the last 40 min, up to 5 min prior to the occurrence of IDH. In particular, many cases were found that exceeded the set maximum area value of upper and lower HRs (=6000 mmHg). In the case of patients with IDH, it can be seen that the maximum area curve of the upper HR (red line) and the maximum curve of the lower HR (green line) gradually increase and reach near the set maximum HR area value.

### 3.4. Perceptron Application

Figure 8 shows the proposed multilayer perceptron structure that uses as input the maximum areas of upper and lower HRs for each of the eight time intervals up to 5 min before IDH occurrence for IDH prediction. The changes in the maximum areas of the lower HRs over time intervals prior to IDH occurrence reflect the decrease in blood pressure due to hypovolemia caused by IDH, while the changes in the maximum areas of the upper HRs indicate the rise in blood pressure resulting from compensatory mechanisms in response to the reduced blood pressure. The proposed multilayer perceptron structure was designed using the Deep Learning Toolbox of MATLAB R2023b. The structure consists of 16 input neurons, 16 neurons per hidden layer with 2 to 5 hidden layers, and 2 neurons in the output layer. Additionally, the scaled conjugate gradient method and cross-entropy cost function were used, with typically 10 to 20 epochs employed.

The performance of the proposed model was evaluated in terms of accuracy (ACC), true positive rate (TPR), positive predictive value (PPV), and Matthews correlation coefficient (MCC), which are calculated as true positive (TP), true negative (TN), false positive (FP), and false negative (FN) values. MCC ranges from −1 to 1, where values closer to 1 indicate a higher similarity between predicted and actual values. ACC, TPR, PPV, and MCC are calculated as ACC = (TP + TN)/(TP + FP + TN + FN), TPR = TP/(TP + FN), PPV = TP/(TP + FP), and MCC = {(TP × TN) − (FN × FP)}/sqrt{(TP + FN)(TN + FP)(TP + FP)(TN + FN)} [24].

Figure 9 shows the confusion matrices for training and test data of multilayer perceptron models with different numbers of hidden layers. In each matrix, the first two diagonal cells show the number and percentage of correct classifications from the training. Each row and column correspond to the predicted and actual classes, respectively. The cells outside the main diagonal represent misclassified observations [40]. In the matrix, the bottom column represents recall (true positive rate) and false negative rate, and the right column represents precision (positive predictive value) and false discovery rate. Additionally, the cell in the bottom right corner represents the overall accuracy [40]. In training with two and five layers, the model’s accuracy using 40 min of data up to 5 min before the onset of IDH was 88.6% and 85.2%, respectively.

Table 3 shows the performance of multilayer perceptron models with different numbers of hidden layers. As a result of testing models with different numbers of hidden layers, models with two and five hidden layers showed good performance in ACC, TPR, and PPV values. Additionally, the MCC values of these two models were also close to 1.

## 4. Discussion

### 4.1. Validity of Data Length Considering Dialysis Time

One important consideration in the data used for IDH research is the time difference between the onset of IDH and the last time point of the data used. In this study, data up to 5 min before the onset of IDH were used. To accurately analyze IDH using HR, it is effective to use the complete HR data preceding the onset of IDH or to utilize data with a small time interval. However, this approach contradicts the problem of predicting IDH in advance (see Appendix A). Additionally, in some cases, the increase in HR due to compensatory mechanisms may occur even after the onset of IDH. Therefore, using the HR signals up to 5 min before the onset of IDH is appropriate; however, in this case, some information regarding the increase in HR (which occurs as a compensatory mechanism) may be lost. On the other hand, using HR signals from 8 or 10 min before the onset of IDH is beneficial for proactive prediction of IDH; however, it has the disadvantage of losing a significant amount of information regarding compensatory mechanisms before the onset of IDH.

The second consideration is the length of the data used. In this study, a data length of 40 min, up to 5 min before the onset of IDH, was used. Longer data lengths facilitate IDH analysis; however, since dialysis sessions typically last 1 to 2 h, the data length cannot exceed this duration. Additionally, longer data lengths make it difficult to predict IDH that occurs early in the dialysis session. Therefore, the length of the data and the accuracy of IDH prediction are in a trade-off relationship with the performance of IDH warning.

### 4.2. Warning Level of IDH Indicator

In this study, the ratio of the maximum sum of changes in the upper and lower HR indicators for most IDH cases exceeded 83%. In contrast, for most normal cases, the ratio of the maximum sum of changes in the upper and lower HR indicators was less than 46%. The issue with the IDH indicators in this study is that the IDH warning threshold (83%) may serve as an urgent alert just before the onset of IDH for some patients, while for others, it may merely be a warning-level alert. To enhance the accuracy of the developed IDH indicators, the current fixed maximum setting value of the sum of the upper and lower HRs of the IDH indicators (=6000) should be personalized for each patient through various experiments.

As shown in the results of the IDH indicators for IDH cases in Figure 6, the maximum area of the upper HR was greater than the maximum area of the lower HR in most cases prior to the onset of IDH. This is a compensatory mechanism for blood pressure elevation in response to hypotension, which indicates that the ratio of the maximum area of the upper HR can be used as a key factor in predicting the onset of IDH. Therefore, in the future, the maximum area settings for the upper and lower HRs that are currently set identically may need to be adjusted differently. Additionally, research is needed to investigate the impact of the duration and kurtosis of the time interval over which the maximum area of the upper HR is calculated on the occurrence of IDH (see Appendix A).

### 4.3. Issue of Varying HR Changes Among Patients

The visual indicator proposed in this paper calculates the maximum area of the upper and lower HRs based on the average HR, accumulating every 5 min from the start of HR measurement, in order to respond to the varying HR changes of each patient. However, the increase and decrease in HR may have limits for each patient. This problem is also related to the warning level of the IDH indicator, as also mentioned in Section 4.2. To address this issue, an IDH indicator can be developed based on the rate of change per unit time for the maximum area of the upper and lower HRs.

### 4.4. Efficiency of Indicator and Perceptron

The perceptron studied in Section 3.2 for IDH prediction may be more useful than the IDH indicators in terms of accuracy for differentiating IDH. However, the IDH indicators have the advantage of allowing real-time monitoring of the patient’s HR and blood pressure status, as well as the progression of IDH. Therefore, utilizing the HR curve, IDH indicators, and perceptron model together may be more effective for the prevention of IDH.

## Figures and Tables

**Figure 1 diagnostics-14-02664-f001:**
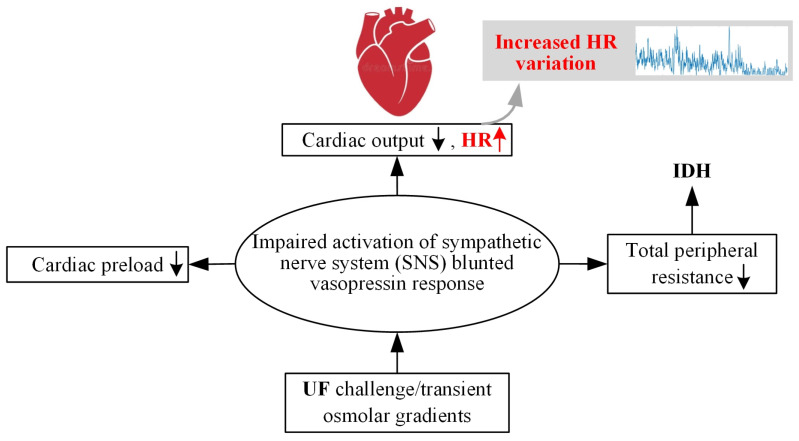
Physiological compensatory response to hypotension that occurs during hemodialysis.

**Figure 2 diagnostics-14-02664-f002:**
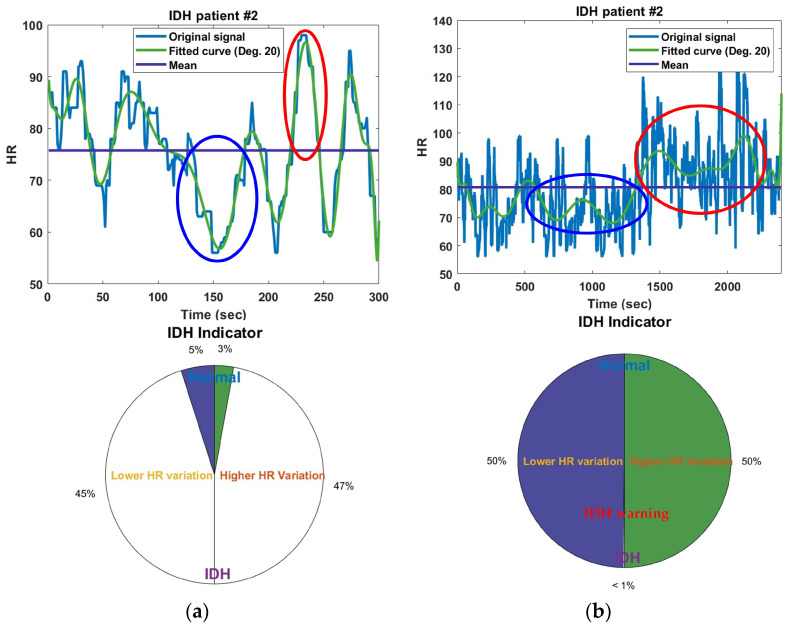
HR signals for (**a**) the first 5 min and (**b**) the last 40 min up to 5 min before IDH occurrence in IDH patients, along with the corresponding IDH indicators. The blue circle (or red circle) represents the area where the local sum of HRs lower (or higher) than the average HR is largest.

**Figure 3 diagnostics-14-02664-f003:**
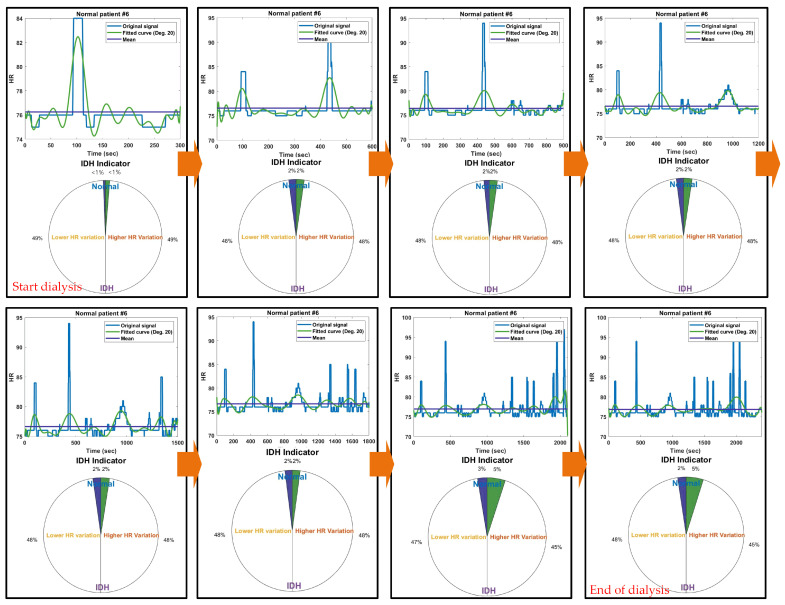
IDH indicator results for each time period for specific normal patients.

**Figure 4 diagnostics-14-02664-f004:**
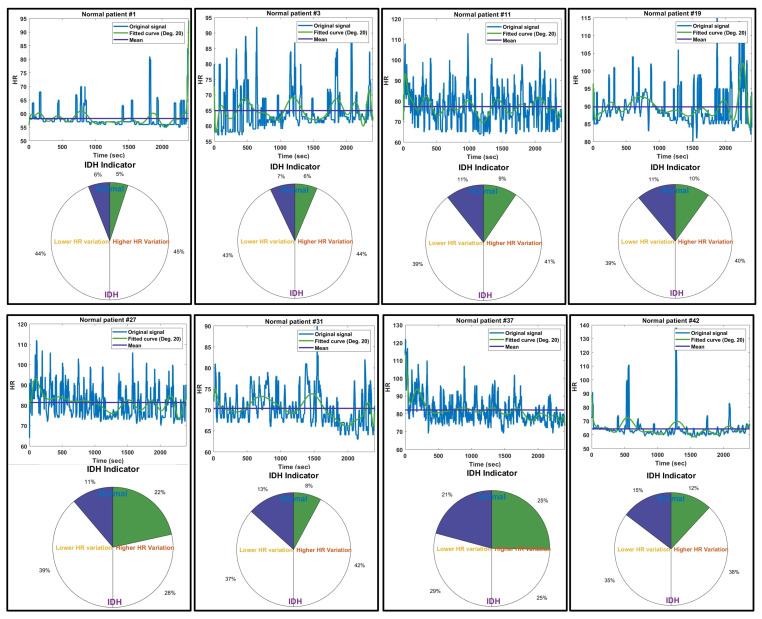
Indicator results for a representative group of eight normal patients who did not experience IDH.

**Figure 5 diagnostics-14-02664-f005:**
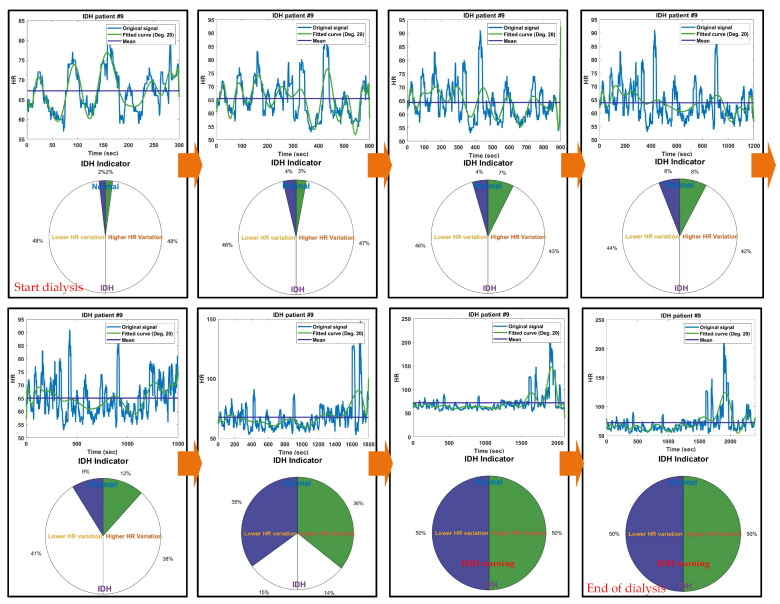
IDH indicator results for each time interval for a specific IDH patient.

**Figure 6 diagnostics-14-02664-f006:**
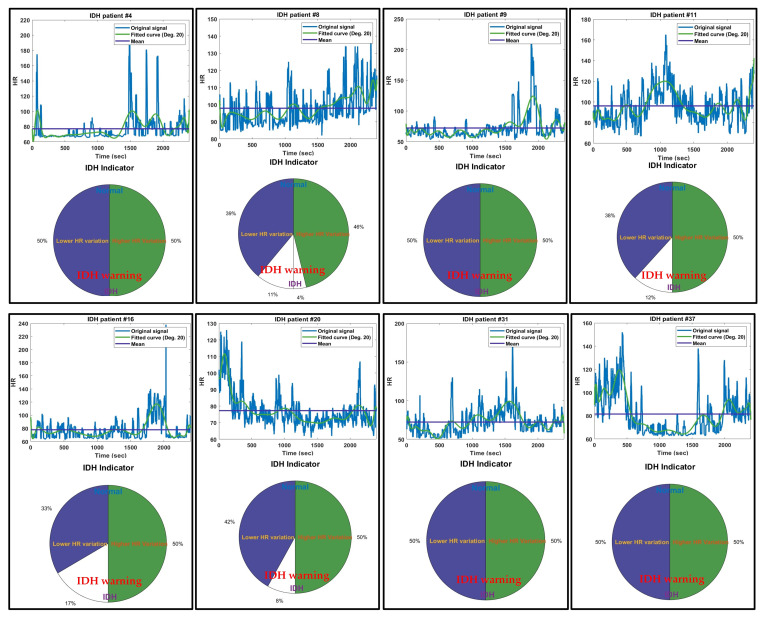
Indicator results for a representative group of eight patients who experienced IDH.

**Figure 7 diagnostics-14-02664-f007:**
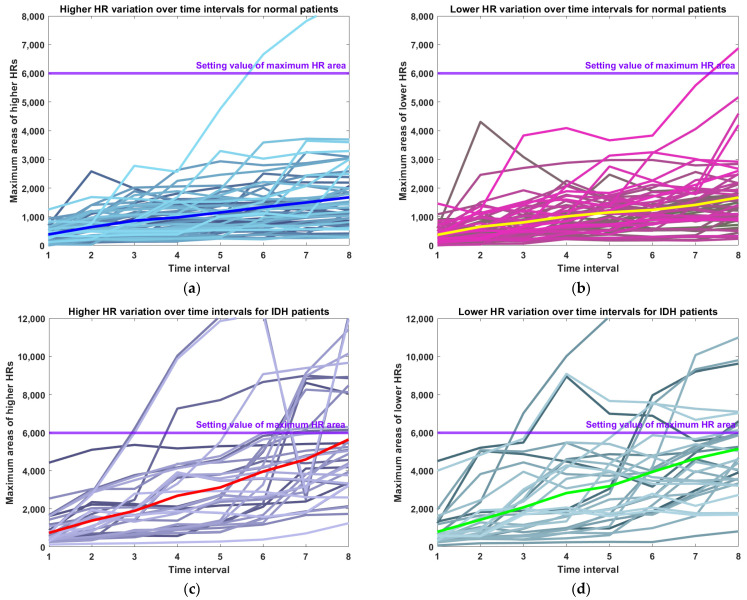
The maximum areas of (**a**) the upper and (**b**) the lower HRs over time intervals for normal patients. The maximum areas of (**c**) the upper and (**d**) the lower HRs over time intervals for IDH patients. The blue (red) and yellow (green) lines represent the average area of high or low HRs relative to the average HR for normal (IDH) patients.

**Figure 8 diagnostics-14-02664-f008:**
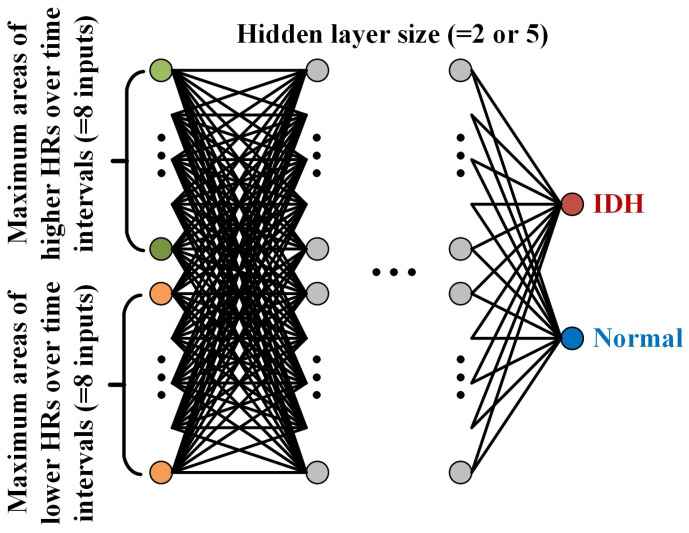
Proposed multilayer perceptron structure for predicting IDH.

**Figure 9 diagnostics-14-02664-f009:**
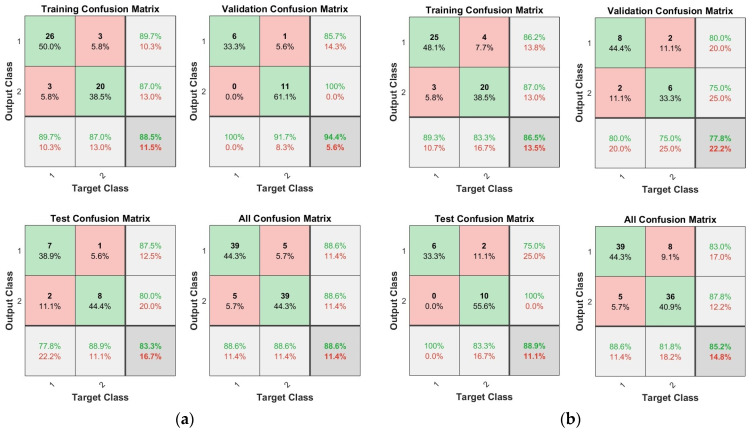
Confusion matrix of multilayer perceptron models according to the number of hidden layers: (**a**) Two layers (88.6%, 11.4%); (**b**) five layers (85.2%, 14.8%). Green cells on the main diagonal correspond to correctly classified observations, and red cells outside the main diagonal correspond to misclassified observations.

**Table 1 diagnostics-14-02664-t001:** Clinical comparison of IDH occurrence group and non-occurrence group.

	Occurrence Group (*n* = 19)	Non-Occurrence Group (*n* = 21)	*p* Value
Age	64.4 ± 10.8	69.1 ± 11.5	0.11
Female (%)	7 (35%)	1 (5%)	0.015
Diabetes	13	9	0.29
Hemoglobin (g/dL)	10.4 ± 0.1	10.2 ± 0.7	0.196
UF amount (kg)	2.65 ± 0.79	2.42 ± 1.05	0.19

**Table 2 diagnostics-14-02664-t002:** The average of the maximum areas of the upper and lower HRs over time intervals for IDH and normal patients.

	5 min	10 min	15 min	20 min	25 min	30 min	35 min	40 min
Upper HR variation for Normal	379.5991	638.9006	860.6037	975.1489	1147.24	1331.16	1496.005	1677.268
Lower HR variation for Normal	367.9417	650.9886	802.9613	1010.646	1150.35	1236.38	1411.816	1668.84
Upper HR variation for IDH	825.661	1477.035	1950.916	2739.734	3182.923	4010.57	4681.352	5546.794
Lower HR variation for IDH	671.2035	1291.615	1949.163	2726.722	3154.916	3917.041	4657.922	5119.467

**Table 3 diagnostics-14-02664-t003:** Performance of multilayer perceptron models according to the number of hidden layers.

Number of Layers	ACC (%)	TPR (%)	PPV (%)	MCC
2 layers	88.6%	88.6%	88.6%	0.773
5 layers	85.2%	83.0%	88.6%	0.706

## Data Availability

The raw data supporting the conclusions of this article will be made available by the authors upon request.

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
