# Peer review of "Visual Indicator for Intradialytic Hypotension Prediction Using Variation and Compensation of Heart Rate"

_diagnostics, 2024, doi:10.3390/diagnostics14232664_

Round 1
Reviewer 1 Report
Comments and Suggestions for Authors
The topic is topical, and, according to the reviewer, it will certainly arouse interest among specialists, and the article will be cited. The analytical aspect of the discussion, the presentation of statistics, and the justified illustrations significantly increase the value of the work. Recommended for publication.
Author Response
<Comments and Suggestions for Authors>
The topic is topical, and, according to the reviewer, it will certainly arouse interest among specialists, and the article will be cited. The analytical aspect of the discussion, the presentation of statistics, and the justified illustrations significantly increase the value of the work. Recommended for publication.
<Answer>
We sincerely appreciate your time and effort in reviewing the paper.
Thank you !!

Reviewer 2 Report
Comments and Suggestions for Authors
Congratulations for your work. I find your paper useful for understanding hemodynamics in dialized patients. Maybe for the future it will be interesting to explore the relationship between heart rate variability in time domain analysis and IDH. In my opinion, it is useful to present your conclusions in a much clear way.
Author Response
<Comments and Suggestions for Authors>
Congratulations for your work. I find your paper useful for understanding hemodynamics in dialized patients. Maybe for the future it will be interesting to explore the relationship between heart rate variability in time domain analysis and IDH. In my opinion, it is useful to present your conclusions in a much clear way.
<Answer>
We sincerely appreciate your time and effort in reviewing the paper.
Thank you !!

Reviewer 3 Report
Comments and Suggestions for Authors
Among several definitions of intradialytic hypotension, Authors should clearly report which one they have adopted in their study, and why.
I suppose each patient has an own cutoff of heart rate increase. So, I’m just wondering how this study could impact on the clinical practice without a specific device which consider a trend in heart rate values and several other parameters.
The article, in the present form, is difficult to follow/understand for a reader, in particular the figures 4-5-6-7-8-10-11-12 are difficult to interpret, and it is not clear how the data can be used concretely in clinical practice.
Authors should reduce the number of the figures, and put not essential figures in supplementary material.
The following reference should be included. This manuscript arrives to similar conclusions, although the approach was significantly different.
Chang YM, Shiao CC, Chang KC, Chen IL, Yang CL, Leu SC, Su HL, Kao JL, Tsai SC, Jhen RN. Heart rate variability is an indicator for intradialytic hypotension among chronic hemodialysis patients. Clin Exp Nephrol. 2016 Aug;20(4):650-659.
Introduction. Among several techniques able to predict a intradialytic hypotensive episode, both in real-time and objectively, Authors should include:
(1) Relative blood volume monitoring [Gabrielli et al, 2009]
(2) B-line quantification at lung ultrasound [Allinovi et al, 2022].
The following references should be included:
Gabrielli D, Krystal B, Katzarski K, Youssef M, Hachache T, Lopot F, Lasseur C, Gunne T, Draganov B, Wojke R, Gauly A. Improved intradialytic stability during haemodialysis with blood volume-controlled ultrafiltration. J Nephrol. 2009 Mar-Apr;22(2):232-40.
Allinovi M, Palazzini G, Lugli G, Gianassi I, Dallari L, Laudicina S, Gregori M, Rossi F, Giannerini D, Cutruzzulà R, Dervishi E, Biagini M, Cirami CL. Pre-Dialysis B-Line Quantification at Lung Ultrasound Is a Useful Method for Evaluating the Dry Weight and Predicting the Risk of Intradialytic Hypotension. Diagnostics (Basel). 2022 Nov 29;12(12):2990.
Author Response
<Comments and Suggestions for Authors>
1) Among several definitions of intradialytic hypotension, Authors should clearly report which one they have adopted in their study, and why.
<Answer>
Following your advice, we have added the explanation below. (line 136, section 2.1)
It was assumed that IDH occurs due to intravascular hypovolemia caused by a de-crease in body fluid during dialysis. Blood pressure lowered by IDH causes a decrease in HR, and a rapid increase in HR was discovered as a compensation mechanism to raise the lowered blood pressure. did. In other words, this paper infers blood pressure changes and IDH through HR changes and proposes a system that informs medical staff of this information in real time. (line 95, Introduction)
In this study, IDH was defined as occurring if systolic blood pressure decreased by more than 20 mmHg even once during dialysis. (line 136, section 2.1)
2) I suppose each patient has an own cutoff of heart rate increase. So, I’m just wondering how this study could impact on the clinical practice without a specific device which consider a trend in heart rate values and several other parameters.
<Answer>
Following your advice, we have added the explanation below to Section 4.3.
4.3. Issue of varying heart rate changes among patients (line 341, Discussion section)
The visual indicator proposed in this paper calculates the maximum area of the upper and lower HRs based on the average HR, accumulating every 5 minutes from the start of HR measurement, in order to respond to the varying HR changes of each patient. However, the increase and decrease in HR may have limits for each patient. This problem is also related to the warning level of the IDH indicator, as also mentioned in Section 4.2. To address this issue, an IDH indicator can be developed based on the rate of change per unit time for the maximum area of the upper and lower HRs.
3) The article, in the present form, is difficult to follow/understand for a reader, in particular the figures 4-5-6-7-8-10-11-12 are difficult to interpret, and it is not clear how the data can be used concretely in clinical practice.
<Answer>
The order of the pictures seems to have confused the reader. Following your advice, I modified the figure and section order as shown below.
3.1. HR changes in Indicators for Normal Patients
3.2. HR changes in Indicators for IDH Patients
3.3. Changes in Maximum Area of Upper and Lower HR for Normal and IDH Patients
3.4. Perceptron Application
4) Authors should reduce the number of the figures, and put not essential figures in supplementary material.
<Answer>
We deleted the existing Figure 2 and combined the descriptions of the existing Figures 2 and 3. Additionally, the existing Figures 11 and 12 have been separated into supplementary figures as shown below.
Supplementary Figure S1. HR signal including compensatory mechanisms for patient with IDH.
Using the heart rate signal up to 3 minutes before the onset of IDH provides the advantage of incorporating the heart rate increase (which occurs as a compensatory mechanism) into the IDH analysis. However, in this case, the warning time for IDH occurrence is not proactive, which is a disadvantage for responding to IDH effectively.
Supplementary Figure S2. Various duration and peaking issues in the time interval in which the maximum HR area is calculated.
IDH cases may have short duration and high kurtosis (Supplementary Figure S2A) or long duration and low kurtosis (Supplementary Figure S2A) in the time interval over which the maximum HR area is calculated.
5) The following reference should be included. This manuscript arrives to similar conclusions, although the approach was significantly different.
Chang YM, Shiao CC, Chang KC, Chen IL, Yang CL, Leu SC, Su HL, Kao JL, Tsai SC, Jhen RN. Heart rate variability is an indicator for intradialytic hypotension among chronic hemodialysis patients. Clin Exp Nephrol. 2016 Aug;20(4):650-659.
Introduction. Among several techniques able to predict a intradialytic hypotensive episode, both in real-time and objectively, Authors should include:
(1) Relative blood volume monitoring [Gabrielli et al, 2009]
(2) B-line quantification at lung ultrasound [Allinovi et al, 2022].
The following references should be included:
Gabrielli D, Krystal B, Katzarski K, Youssef M, Hachache T, Lopot F, Lasseur C, Gunne T, Draganov B, Wojke R, Gauly A. Improved intradialytic stability during haemodialysis with blood volume-controlled ultrafiltration. J Nephrol. 2009 Mar-Apr;22(2):232-40.
Allinovi M, Palazzini G, Lugli G, Gianassi I, Dallari L, Laudicina S, Gregori M, Rossi F, Giannerini D, Cutruzzulà R, Dervishi E, Biagini M, Cirami CL. Pre-Dialysis B-Line Quantification at Lung Ultrasound Is a Useful Method for Evaluating the Dry Weight and Predicting the Risk of Intradialytic Hypotension. Diagnostics (Basel). 2022 Nov 29;12(12):2990.
<Answer>
We have included explanations of the references you provided in the Introduction as shown below. (line 65, Introduction)
The currently proposed IDH predictors or indicators have used heart rate variability (HRV) [17], UF rate control based on blood volume [18,19], and fluid status assessment based on B-line count [20] as representative factors for predicting IDH. In [17], it was found that patients who developed IDH had hypertension at the start of dialysis, and their HRV indices decreased as the dialysis progressed. However, to ensure the reliability of the results, more HRV measurements and corresponding blood pressure measurements during dialysis are required, but obtaining such data during dialysis is practically challenging. In [18], IDH was analyzed to be caused by hypovolemia resulting from UF, and it was reported that blood volume (BV) control through UF contributes to intra-dialytic stability. On the other hand, [19] reported that dialysis using BV change-guided UF control did not reduce the prevalence of IDH compared to standard dialysis, contrary to the findings of [18]. In [20], it was reported that arrhythmias, low pre-dialysis systolic blood pressure, and a low number of B-lines on lung ultrasound were associated with a patient's fluid status and the development of IDH. However, due to the small number of clinical patients, it remains unclear whether the routine use of lung ultrasound is effective in preventing IDH. (line 65, Introduction)
We sincerely appreciate your time and effort in reviewing the paper. Thank you !!

Round 2
Reviewer 3 Report
Comments and Suggestions for Authors
Authors claim: “IDH was defined as occurring if systolic blood pressure decreased by more than 20 mmHg even once during dialysis”. Even those asymptomatic episodes? This should be specified.
is the adopted definition of IDH widely accepted?
Author Response
<Comments and Suggestions for Authors>
Authors claim: “IDH was defined as occurring if systolic blood pressure decreased by more than 20 mmHg even once during dialysis”. Even those asymptomatic episodes? This should be specified.
is the adopted definition of IDH widely accepted?
<Answer>
Yes, that's correct. As defined in line 32 of the Introduction, the definition of IDH is widely accepted. Clinical data analysis revealed that patients with IDH frequently experienced systolic blood pressure drops of more than 20 mmHg on multiple occasions. Therefore, even a single instance of a systolic blood pressure drop exceeding 20 mmHg was considered an occurrence of IDH, as it could indicate a potentially dangerous condition. However, such cases were not very common in the clinical data.
We sincerely appreciate your time and effort in reviewing the paper.
Thank you !!
